# Representation Learning of Compositional Data

**Marta Avalos-Fernandez**[†]  **Richard Nock**[‡§♮]  **Cheng Soon Ong**[‡§]  **Julien Rouar**[†]  **Ke Sun**[‡] *

[†]Université de Bordeaux,  [‡]Data61,
[§]the Australian National University and [♮]the University of Sydney
`first.last@{u-bordeaux.fr,data61.csiro.au}`

## Abstract

We consider the problem of learning a low dimensional representation for compositional data. Compositional data consists of a collection of nonnegative data that sum to a constant value. Since the parts of the collection are statistically dependent, many standard tools cannot be directly applied. Instead, compositional data must be first transformed before analysis. Focusing on principal component analysis (PCA), we propose an approach that allows low dimensional representation learning directly from the original data. Our approach combines the benefits of the log-ratio transformation from compositional data analysis and exponential family PCA. A key tool in its derivation is a generalization of the scaled Bregman theorem, that relates the perspective transform of a Bregman divergence to the Bregman divergence of a perspective transform and a *remainder* conformal divergence. Our proposed approach includes a convenient surrogate (upper bound) loss of the exponential family PCA which has an easy to optimize form. We also derive the corresponding form for nonlinear autoencoders. Experiments on simulated data and microbiome data show the promise of our method.

## 1  Introduction

Compositional data analysis (CoDA) is a subfield of statistics introduced more than three decades ago [3, 2, 1, 29]. Compositional data consist of a collection of nonnegative measurements that sum to a constant value, typically, proportions that sum to 1. Because knowing the sum, one component can be determined from the sum of the remainder, the parts that make up the composition are mathematically and statistically dependent. This distinct structure complicates analysis and does not allow standard statistical analyses. Ignoring the underlying nature of the data studied might give rise to misleading conclusions.

Among others, [1] and [13] provided a framework to perform CoDA by mapping data from the constrained simplex space to the Euclidian space using nonlinear log-ratio transforms. In this paper, we focus on Principal Components Analysis (PCA), one of the main tools for exploratory analysis of compositional data. Just like in standard Euclidean data, it is particularly useful when the first few principal components explain enough variability to be considered as representative. Unfortunately, any operation of centering or scaling destroys the compositional nature of the data, which complicates a direct application of PCA.

Our motivation for studying CoDA comes from the recent explosion of microbiome studies [14, 15]. Indeed, spectacular advances in 16S rRNA gene sequencing of the bacterial component of the human microbial community (microbiota) have enabled researchers to investigate human health and disease, leading to new insights into the role of these microbial communities. The microbiota sequencing data are measured as read counts interpreted as a species' abundance in a microbial community. To make the microbial abundance comparable across samples, data are normalized to the relative abundances

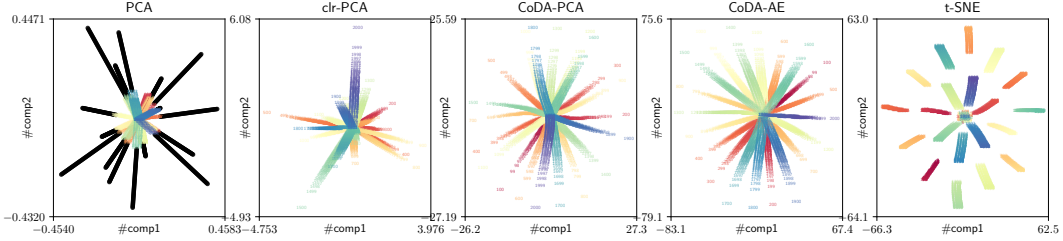

Figure 1: 2D Visualization of the low dimensional representation A on the `arms` dataset.

of all bacteria observed. On the other hand, because high-throughput experiments produce large amounts of data, multivariate analysis is indispensable [27, 21]. There is a stress to understand the soundness of models [5].

In this paper, we propose to learn a low dimensional representation of CoDA from the *original* data. To account for the nonlinearity due to the compositional nature of the data, we start from exponential family PCA [12] that we augment with the compositional constraint and then simplify the loss to be optimized via a generalization of a recent result [25] on Bregman divergences, which may be of independent interest. We also propose a nonlinear autoencoder (AE) version to learn the low dimensional representation.

Let us examine a toy example to illustrate our approach. We generate the `arms` dataset in $\mathfrak{S}^{19}$ by evenly interpolating the simplex center and each of the 20 vertices with 100 points, therefore yielding a matrix $\mathsf{X}_{20 \times 2000}$. Figure 1 shows the 2D representation $\mathsf{A}_{2 \times 2000}$ computed by five methods: the standard PCA; `clr-PCA` computes the standard PCA after performing clr; `CoDA-PCA` and `CoDA-AE` are our proposed methods; t-SNE [32] is a popular nonlinear dimensionality reduction method and is applied on $\mathsf{X}$ directly. In the `PCA` plot, the black segments indicate that the PCA reconstruction is outside of the simplex. PCA cannot be directly adapted to CoDA because the projection on the principal components may go beyond the convex hull of the vertices. It is clear that only `CoDA-PCA` and `CoDA-AE` uncover the true structure, where all the arms are clearly presented, and their connections are faithfully presented.

## 2 Compositional Data Analysis

We briefly review some definitions of CoDA. Compositional data are proportions: $\mathsf{X}$ is a compositional dataset if and only if $\mathsf{X} \in \Re^{d \times m}$ such that $\forall i \in [1, ..., m]$ the vector column $\boldsymbol{x}_i$ of $\mathsf{X}$ is in the simplex $\mathfrak{S}^d = \left\{ \boldsymbol{x} \in \Re^d \; : \; \forall j, x_j > 0; \; \sum_{j=1}^{d} x_j = \kappa \right\}$, where $\kappa > 0$ is a constant, classically 1. Here the superscript $d$ does not denote the dimensionality as $\dim(\mathfrak{S}^d) = d - 1$. For a dataset $\mathsf{X}'$ which contains counts of strictly positive values, we reduce it to a compositional dataset by dividing out the totals, that is we compute the CoDA set $\mathsf{X}$ such that: $\boldsymbol{x}_i = \boldsymbol{x}'_i \frac{1}{\sum_{j=1}^{d} x'_{ji}}$ is the vector of proportions for individual $i$.

Using Bregman divergences makes explicit a dual affine [6] coordinate space which is in fact the log coordinates of Aitchison [1]. It is in this space that we have affine constraints, which are therefore non-linear in the "primal", ambient space. To manage this nonlinear structure, it has been proposed [4] to first apply a log-ratio transformation to transpose the data into real Euclidean space. For instance, the additive log-ratio transformation (alr) applies log-ratio between a component and a reference component; the centered log-ratio transformation (clr) scales each subject vector by its geometric mean; and the isometric log-ratio transformation (ilr) is associated with an orthogonal coordinate system in the simplex. Afterwards, standard PCA is performed.

By definition, the clr transformation is

$$\boldsymbol{c}_{\text{KL}}(\boldsymbol{x}) = \log\left(\frac{\boldsymbol{x}}{g(\boldsymbol{x})}\right) = \mathsf{C}^{\text{clr}} \log(\boldsymbol{x}) = \log(\boldsymbol{x}) - \overline{\log(\boldsymbol{x})}\mathbf{1}_d, \tag{1}$$

where $g(\boldsymbol{x}) = (\prod_{j=1}^{d} x_j)^{1/d}$ the geometric mean of $\boldsymbol{x}$, $\overline{\boldsymbol{x}} = \frac{1}{d}\sum_{j=1}^{d} x_j$ is the arithmetic mean of $\boldsymbol{x}$, $\mathbf{1}_d$ is the $1 \times d$ vector of all ones, and $\mathsf{C}^{\text{clr}} = \mathsf{I}_d - \frac{1}{d}\mathbf{1}_d\mathbf{1}_d^{\mathsf{T}}$. The purpose of the log-ratio transformation

(centered or not) is to go back to $\Re^d$ from $\mathfrak{S}^d$ without losing information. Notice that $\log(x_j)$, $\overline{\log(\boldsymbol{x})} \in (-\infty, 0)$ so the compositional data is embedded in $\Re^d$ under the clr transformation. The reverse operation: $\boldsymbol{x} = \boldsymbol{c}_{\text{KL}}^{-1}(\boldsymbol{x}') = \exp(\boldsymbol{x}') \prod_{j=1}^{d} \exp(\frac{1}{1-d} x_j')$ embeds $\Re^d$ into $\mathfrak{S}^d$. See the table below for a comparison of clr, alr and ilr, presented as different transformations $\mathsf{C} \log(\boldsymbol{x})$. They are equivalent up to linear transformations. Without loss of generality we focus on the clr.

| clr | alr | ilr |
|---|---|---|
| $\mathsf{C}_{d \times d}^{\text{clr}} = \mathsf{I}_d - \frac{1}{d} \mathbf{1}_d \mathbf{1}_d^{\mathsf{T}}$ | $\mathsf{C}_{(d-1) \times d}^{\text{alr}} = [\mathsf{I}_{d-1}, -\mathbf{1}_{d-1}]$ | $\mathsf{C}_{(d-1) \times d}^{\text{ilr}} \in \{\mathsf{R}\mathsf{C}^{\text{clr}} : \mathsf{R}\mathsf{R}^{\mathsf{T}} = \mathsf{I}_{d-1}\}$ |

However interpreting the resulting coordinates is still challenging [13, 24]: alr transformation is no distance-preserving; clr leads to degenerate distributions and singular covariance matrices; ilr avoids the precedent drawbacks, but still, results from complicated nonlinear transformations are difficult to interpret. Currently, there seems to be no consensus about the best practices ([16] versus [31]) and, in all cases, log-transforming is not a remedy for all the difficulties arisen by CoDA [20].

# 3 Exponential Family Principal Component Analysis

Another way to apply dimension reduction is to perform a generalized PCA on crude count data. Based on the same ideas as the generalised linear model, [12] described a generalized PCA model for distributions from the exponential family. We first recall the standard PCA setting.

## 3.1 Principal Component Analysis

For simplicity suppose that the data matrix $\mathsf{X}$ is already centered that can be easily achieved by appending to $\mathsf{A}$ matrix a row of ones.

---

(Traditional PCA) We have a dataset $\mathsf{X} \in \Re^{d \times m}$ that we approximate as $\mathsf{X} \sim \mathsf{V}^{\mathsf{T}}\mathsf{A}$ by minimizing the following loss wrt the constraints $\mathsf{A} \in \Re^{\ell \times m}$, $\mathsf{V} \in \Re^{\ell \times d}$, $\mathsf{V}\mathsf{V}^{\mathsf{T}} = \mathsf{I}_\ell$:

$$\ell_{\text{PCA}}(\mathsf{X}; \mathsf{A}, \mathsf{V}) \doteq \|\mathsf{X} - \mathsf{V}^{\mathsf{T}}\mathsf{A}\|_{\text{F}}^2 \ . \tag{2}$$

---

Hence, observations are column-wise. $\mathsf{V} : \Re^d \to \Re^\ell$ is surjective with $\mathsf{V}^{\mathsf{T}}\mathsf{V}$ defining a rank-$\ell$ projection, assuming in general $\ell < d$. $\mathsf{A}$ is the representation of data points. The goodness of fit of the representation is measured by the squared Frobenious norm. We summarise the different transformations and loss functions in Table 1.

Observe that instead of finding a linear representation $\mathsf{A}$ and its corresponding linear loadings $\mathsf{V}$, we can consider nonlinear functions for encoding and decoding the latent representation. When the nonlinear encoder and decoder are implemented as feed-forward neural networks, we arrive at the autoencoder setting.

## 3.2 Bregman Divergence and $\varphi$-PCA

As mentioned in the introduction, compositional data do not live in Euclidean space. Count data are naturally linked to the Poisson distribution, and therefore we should consider an exponential family model for count data. From the Bayesian viewpoint, the PCA goal is to minimize a distance ($L^2$ for usual PCA) which is equivalent to a Bregman divergence minimization (or a Likelihood function maximization).

**Definition 1 (Bregman Divergence)** *Let $\varphi : \Re^d \to \Re$ convex differentiable. The Bregman divergence $D_\varphi$ with generator $\varphi$ is*

$$D_\varphi(\boldsymbol{x} \, \| \, \boldsymbol{x}') \doteq \varphi(\boldsymbol{x}) - \varphi(\boldsymbol{x}') - (\boldsymbol{x} - \boldsymbol{x}')^{\mathsf{T}} \nabla \varphi(\boldsymbol{x}'). \tag{3}$$

A Bregman divergence is just a truncation of the Taylor expansion of a function. It can therefore be defined for any differentiable function, not just the convex ones. If $\varphi$ is *not* convex, we call $D_\varphi$ a *Bregman distortion*, which is a signed dissimilarity. We denote by $\varphi^\star(\boldsymbol{x}) \doteq \sup_{\boldsymbol{y}} \{\boldsymbol{x}^{\mathsf{T}}\boldsymbol{y} - \varphi(\boldsymbol{y})\}$ the convex conjugate of the generator $\varphi$ [9].

Table 1: Summary of methods in this paper

| Method | Original | Reconstruction | Distortion | Notes |
|--------|----------|----------------|------------|-------|
| PCA | $\mathsf{X}$ | $\mathsf{V}^\mathsf{T}\mathsf{A}$ | $\\|\cdot - \cdot\\|_\mathrm{F}^2$ | classical PCA(2) |
| $\varphi$-PCA | $\mathsf{X}$ | $\nabla^*\varphi(\mathsf{V}^\mathsf{T}\mathsf{A})$ | $D_\varphi(\cdot\\|\cdot)$ | exponential family PCA (4) |
| clr-PCA | $\boldsymbol{c}_\mathrm{KL}(\mathsf{X})$ | $\mathsf{V}^\mathsf{T}\mathsf{A}$ | $\\|\cdot - \cdot\\|_\mathrm{F}^2$ | CoDA with clr (5) |
| gauged-$\varphi$-PCA | $\nabla\varphi(\check{\mathsf{X}})$ | $\mathsf{V}^\mathsf{T}\mathsf{A}$ | $D_{\varphi^\star}(\cdot\\|\cdot)$ | General Bregman PCA (8) |
| CoDA PCA | $\boldsymbol{c}_\mathrm{KL}(\mathsf{X})$ | $\mathsf{V}^\mathsf{T}\mathsf{A}$ | $D_\mathrm{exp}(\cdot\\|\cdot)$ | (11) is a special case of (8) |
| s-CoDA-PCA | $\check{\boldsymbol{x}}_i$ | $\nabla\check{\mathrm{KL}}(\exp(\mathsf{V}^\mathsf{T}\boldsymbol{a}_i))$ | inner product | upper bound (17) |
| CoDA AE | $\mathsf{X}$ | $g_\theta \circ h_\Phi(\mathsf{X})$ | $D_\mathrm{exp}(\cdot\\|\cdot)$ | neural networks $g_\theta$ and $h_\Phi$ |

PCA has been generalized to the exponential families in a way that makes fitting occur in the *natural parameter* space [12, 19] (and references therein). The optimization problem is non-convex. The algorithmic strategy proposed by [12] is to use an alternating sequence of convex minimizations under constraints. Alternatively, [19] proposed maximizing the deviance (as a generalized notion of variance) and [10] proposed maximizing the likelihood function via a variational algorithm and gradient descent.

We denote exponential family PCA as $\varphi$-PCA, where $\varphi$ is the cumulant of the exponential family, which is strictly convex differentiable with convex conjugate $\varphi^\star$, and uniquely determines the exponential family under mild conditions [7]. Note that for $\varphi$-PCA, $\mathsf{X}$ is not neccessarily in a vector space (e.g. $\mathsf{X} \not\subseteq \Re^{d\times m}$).

> ($\varphi$-PCA) We have a dataset $\mathsf{X}_{d\times m}$ that we approximate as $\mathsf{X} \sim \nabla\varphi^\star(\mathsf{V}^\mathsf{T}\mathsf{A})$ with $\mathsf{A} \in \Re^{\ell\times m}$, $\mathsf{V} \in \Re^{\ell\times d}$, $\mathsf{V}\mathsf{V}^\mathsf{T} = \mathsf{I}_\ell$, through minimizing the Bregman loss
>
> $$\ell_{\varphi\text{-PCA}}(\mathsf{X};\mathsf{A},\mathsf{V}) \doteq \sum_i D_\varphi(\boldsymbol{x}_i \\| \nabla\varphi^\star(\mathsf{V}^\mathsf{T}\boldsymbol{a}_i)) = D_\varphi(\mathsf{X} \\| \nabla\varphi^\star(\mathsf{V}^\mathsf{T}\mathsf{A})). \tag{4}$$

Vectors are column-vectors: $\boldsymbol{x}_i, \boldsymbol{a}_i$ are respectively column observation $i$ in the ambient and principal spaces, respectively. This formulation has a major advantage that linear algebra may be used to fit $\mathsf{A},\mathsf{V}$ while $\mathsf{X}$ may not lie in a vector space, see for example [12, 19] and references therein. We remark that because of the dual symmetry of Bregman divergences, we have $D_\varphi(\mathsf{X} \\| \nabla\varphi^\star(\mathsf{V}^\mathsf{T}\mathsf{A})) = D_{\varphi^\star}(\mathsf{V}^\mathsf{T}\mathsf{A} \\| \nabla\varphi(\mathsf{X}))$ [8]. Notice there exists a little "hole" in the $\varphi$-PCA definition, as $\mathsf{X}$ is not necessarily easy to center when it is not in a vector space.

$\varphi$-PCA includes standard PCA as a special case as when $\varphi(\boldsymbol{x}) = \frac{1}{2}\\|\boldsymbol{x}\\|_\mathrm{F}^2$ and the corresponding Bregman divergence becomes $D_\varphi(\boldsymbol{x} \\| \boldsymbol{x}') = \frac{1}{2}\\|\boldsymbol{x} - \boldsymbol{x}'\\|_\mathrm{F}^2$ .

## 4   Exponential family PCA on Compositional Data

CoDA has found a workaround for the centering problem, *centered log-ratio coordinates*. From [3, Def. 4.6, Chap. 8] the associated loss is the standard PCA loss on clr transformed data:

$$\ell_{\text{clr-PCA}}(\mathsf{X};\mathsf{A},\mathsf{V}) \doteq \frac{1}{2}\\|\boldsymbol{c}_\mathrm{KL}(\mathsf{X}) - \mathsf{V}^\mathsf{T}\mathsf{A}\\|_\mathrm{F}^2 = D_\varphi(\boldsymbol{c}_\mathrm{KL}(\mathsf{X}) \\| \nabla\varphi^\star(\mathsf{V}^\mathsf{T}\mathsf{A})), \tag{5}$$

where $\boldsymbol{c}_\mathrm{KL}(\mathsf{X})$ is the centered log-ratio transform defined in Equation (1) and $\varphi(\boldsymbol{x}) = \frac{1}{2}\\|\boldsymbol{x}\\|_\mathrm{F}^2$. Recall from the previous section that we could deal with crude count data by using exponential family PCA. However if we wish to perform PCA on the crude count data, while maintaining the clr transform, we need an additional normalization term, which requires us to obtain a gauged version of the Bregman divergence.

### 4.1   Scaled Bregman Theorem with Remainder

In this section we generalize the Scaled Bregman Theorem from [25, Theorem 1] to allow for a remainder term. We use it in this paper to deal with the perspective transform required for CoDA, but it may be of independent interest. Recall that $\varphi$ is the generator of the Bregman distortion (Definition 1). We additionally define a perspective (or gauge) function $g$ to deal with the fact that we

are considering data on the simplex. Whenever $\varphi$ and $g$ are differentiable, the following is immediate from [25, Theorem 1].

---

**Theorem 2 (Scaled Bregman Theorem with Remainder)** *For any* $\varphi : \mathfrak{X} \to \Re$ *and* $g : \mathfrak{X} \to \Re_*$ *($\Re_* = \Re \setminus \{0\}$) that are both differentiable, denoting*

$$\check{\boldsymbol{x}} \doteq \frac{\boldsymbol{x}}{g(\boldsymbol{x})} \quad and \quad \check{\varphi}(\boldsymbol{x}) \doteq g(\boldsymbol{x}) \cdot \varphi\left(\frac{\boldsymbol{x}}{g(\boldsymbol{x})}\right) \ , \tag{6}$$

*the following holds true:*

$$g(\boldsymbol{x}) \cdot D_\varphi\left(\check{\boldsymbol{x}} \,\|\, \check{\boldsymbol{y}}\right) = D_{\check{\varphi}}\left(\boldsymbol{x} \,\|\, \boldsymbol{y}\right) + R_{\varphi,g}(\boldsymbol{x} \,\|\, \boldsymbol{y}) \ , \quad \forall \boldsymbol{x}, \boldsymbol{y} \in \mathfrak{X} \ , \tag{7}$$

*where* $R_{\varphi,g}(\boldsymbol{x} \,\|\, \boldsymbol{y}) \doteq \varphi^\star\left(\nabla\varphi(\check{\boldsymbol{y}})\right) \cdot D_g(\boldsymbol{x} \,\|\, \boldsymbol{y})$ *is called the* remainder.

---

We can abstract Theorem 2 by saying that for any $\varphi, g$ differentiable, we have

$$\text{perspective-Bregman}(\varphi, g) = \text{Bregman}(\text{perspective}(\varphi)) + \text{conformal-Bregman}(g, \varphi),$$

where "perspective$(\varphi)$" is $\check{\varphi}$ in (6), and conformal divergences are defined and analyzed in [26]. General classes of perspective transforms of convex functions are introduced in [22, 23]. The notion of perspective transform of a Bregman divergence was introduced in [25]. In [25, Theorem 1], conditions are assumed that make $R_{\varphi,g}(\boldsymbol{x} \,\|\, \boldsymbol{y}) = 0$, resulting in the scaled Bregman theorem. Notice that $D_\varphi$ is a Bregman distortion but not necessarily a Bregman divergence if $\varphi$ is not convex. For reasons explained in [25], we call $g$ a *gauge*. In the following we assume that $\varphi$ is separable, so that we can use both notations $\nabla\varphi$ and $\varphi'$ to denote the gradient and derivatives involving $\varphi$.

By Theorem 2, as long as $g(\boldsymbol{x})$ is homogeneous of degree one, $D_\varphi\left(\check{\boldsymbol{x}} \,\|\, \check{\boldsymbol{y}}\right)$ and $\frac{1}{g(\boldsymbol{x})}\left[D_{\check{\varphi}}\left(\boldsymbol{x} \,\|\, \boldsymbol{y}\right) + R_{\varphi,g}(\boldsymbol{x} \,\|\, \boldsymbol{y})\right]$ are both invariant to re-scaling of $\boldsymbol{x}$ and $\boldsymbol{y}$ and can therefore be used to deal with compositional data. A general formulation of $g$ satisfying this condition can be $g(\boldsymbol{x}) = \prod_{j=1}^d x_j^{w_j}$, where $\forall j, w_j \geq 0$ and $\sum_{j=1}^d w_j = 1$. In this paper, we focus on the special case $\forall j, w_j = \frac{1}{d}$ so that $D_\varphi\left(\check{\boldsymbol{x}} \,\|\, \check{\boldsymbol{y}}\right)$ can be expressed in terms of the widely used clr transformation. Setting $\boldsymbol{w}$ to be a one-hot vector $(1, 0, \cdots, 0)$ can express $D_\varphi\left(\check{\boldsymbol{x}} \,\|\, \check{\boldsymbol{y}}\right)$ with the alr. This latter case will be omitted here.

## 4.2 Exponential Family CoDA

We are now in a position to derive the exponential family version of the loss in (5). Let $\check{X}$ denote the matrix of the column vectors $\check{\boldsymbol{x}}_i$. It turns out that in the same way as (2) is an approximation of (4), the loss in (5) is an approximation of the gauged loss:

$$\ell_{\text{gauged-}\varphi\text{-PCA}}(X; A, V) \doteq D_{\varphi^\star}(V^\mathsf{T}A \,\|\, \nabla\varphi(\check{X})) = D_\varphi(\check{X} \,\|\, \nabla\varphi^\star(V^\mathsf{T}A)). \tag{8}$$

Note that the above expression is in terms of the normalised matrix $\check{X}$. To unpack it in terms of the original data $X$, we apply Theorem 2. In the CoDA case, $\varphi^\star(z) \doteq \exp z$, the convex dual of $\varphi(z) \doteq z \log z - z$. Indeed, after remarking that $\nabla\varphi(\check{X}) = \boldsymbol{c}_{\text{KL}}(X)$, it follows

$$\ell_{\text{gauged-KL-PCA}}(X; A, V) = D_{\exp}(V^\mathsf{T}A \,\|\, \boldsymbol{c}_{\text{KL}}(X)) = D_{\text{KL}}\left(\check{X} \,\|\, \exp(V^\mathsf{T}A)\right)$$

$$= \mathbf{1}^\mathsf{T}\exp(V^\mathsf{T}A)\mathbf{1} - \text{trace}\left(\check{X}^\mathsf{T}V^\mathsf{T}A\right) + \text{constant}. \tag{9}$$

In other words, the CoDA PCA is in fact fitting natural parameters from centered log-ratios being natural coordinates as well. From (9) we observe that both of them live in the same space. Therefore $V^\mathsf{T}A$ is centered in the same way as $\boldsymbol{c}_{\text{KL}}(X)$, and so

$$V\mathbf{1}_d \in \ker(A^\mathsf{T}) \quad \Leftrightarrow \quad A^\mathsf{T}V\mathbf{1} = \mathbf{0}_m \ . \tag{10}$$

Remark that a centering assumption is also explicit in [3, Chapter 8, Eq. 8.1].

Hence, we can define the CoDA PCA problem as follows.

(CoDA PCA) We have a dataset $\mathsf{X} \in (\mathfrak{S}^d)^m$ that we approximate as $\boldsymbol{c}_{\text{KL}}(\mathsf{X}) \sim \mathsf{V}^\intercal \mathsf{A}$ by minimizing the following loss wrt the constraints $\mathsf{A} \in \Re^{\ell \times m}$, $\mathsf{V} \in \Re^{\ell \times d}$, $\mathsf{V}\mathsf{V}^\intercal = \mathsf{I}_\ell$, $\mathsf{A}^\intercal \mathsf{V}\mathbf{1} = \mathbf{0}$:

$$\ell_{\text{CoDA-PCA}}(\mathsf{X}; \mathsf{A}, \mathsf{V}) = D_{\exp}(\mathsf{V}^\intercal \mathsf{A} \,\|\, \boldsymbol{c}_{\text{KL}}(\mathsf{X})). \tag{11}$$

Regarding $\boldsymbol{c}_{\text{KL}}(\boldsymbol{x})$, $\check{\boldsymbol{x}}$ and $\boldsymbol{x}$ as different coordinate systems of $\mathfrak{S}^d$, we use the Fisher information metric (FIM) [6], whose formulation is well studied on the $\boldsymbol{x}$ coordinates, to define the corresponding *pullback metric* $\mathcal{G}$ under the $\boldsymbol{c}_{\text{KL}}(\boldsymbol{x})$ and $\check{\boldsymbol{x}}$ coordinates, meaning that these metrics correspond to the same underlying geometry of $\mathfrak{S}^d$. We have the following proposition (proof omitted; see [30] for similar derivations).

**Proposition 3** *The FIM that uniquely defines the geometry of $\boldsymbol{c} \in \{\boldsymbol{c}_{\text{KL}}(\boldsymbol{x}) : \boldsymbol{x} \in \mathfrak{S}^d\}$ is given by* $\mathcal{G}_{ij}(\boldsymbol{c}) = \delta_{ij} \frac{\exp(c_i)}{\sum_{i=1}^d \exp(c_i)} - \frac{\exp(c_i + c_j)}{\left(\sum_{i=1}^d \exp(c_i)\right)^2}$; *the FIM under the coordinates $\check{\boldsymbol{x}}$ is given by* $\mathcal{G}_{ij}(\check{\boldsymbol{x}}) = \delta_{ij} \frac{1}{\check{x}_i \sum_{i=1}^d \check{x}_i} - \frac{1}{(\sum_{i=1}^d \check{x}_i)^2}$, *where $\delta_{ij} = 1$ if $i = j$ otherwise $\delta_{ij} = 0$.*

Intuitively, the metric $\mathcal{G}$ measures the local distance $\mathrm{d}\check{\boldsymbol{x}}^\intercal \mathcal{G}(\check{\boldsymbol{x}})\mathrm{d}\check{\boldsymbol{x}}^\intercal$ of a tiny shift $\mathrm{d}\check{\boldsymbol{x}}$. It is not everywhere identity as in a Euclidean space. Therefore the distance should not be measured by the Frobenious norm as in (5). In contrast, our loss $\ell_{\text{CoDA-PCA}}(\mathsf{X}; \mathsf{A}, \mathsf{V})$ is based on the KL divergence which locally agrees with the FIM [6].

### 4.3 Relating CoDA PCA to $\varphi$-PCA

We now define and analyze a generalized perspective transform of the generator of KL divergence: let $\check{\text{KL}}(\boldsymbol{x}) \doteq g(\boldsymbol{x}) \cdot \sum_{j=1}^d \varphi(x_j/g(\boldsymbol{x}))$ where $\varphi(z) \doteq z \log(z) - z$ and $g(\boldsymbol{x}) \doteq (\prod_j x_j)^{1/d}$.

**Lemma 4** *(Properties of $\check{\text{KL}}$) $\check{\text{KL}}$ satisfies the following properties:*

*(1) $\check{\text{KL}}$ is convex;*

*(2) the general term of the Hessian $\mathsf{H}$ of $\check{\text{KL}}$ is*

$$\mathsf{H}_{ij} \doteq \mathsf{H}_{ij}(\check{\text{KL}}(\boldsymbol{x})) = \frac{1}{dx_j} \cdot \begin{cases} -u_{ji} & if \quad j \neq i \\ \sum_{k \neq j} u_{kj} & otherwise \end{cases}, \tag{12}$$

*where $u_{ab} \doteq 1 + x_a/x_b$. Furthermore,*

$$\boldsymbol{z}^\intercal \mathsf{H}\boldsymbol{z} = \frac{1}{2d} \sum_{ij} (x_i + x_j) \cdot \left(\frac{z_i}{x_i} - \frac{z_j}{x_j}\right)^2, \forall \boldsymbol{z} \in \Re^d. \tag{13}$$

*Hence, $\boldsymbol{z}^\intercal \mathsf{H}\boldsymbol{z} \geq 0, \forall \boldsymbol{x} \in \Re_{++}^d, \forall \boldsymbol{z} \in \Re^d$ and $\boldsymbol{z}^\intercal \mathsf{H}\boldsymbol{z} = 0$ only when $\boldsymbol{z} \propto \boldsymbol{x}$;*

*(3) function $\check{\text{KL}} \circ \exp$ is 1-homogeneous on $\text{span}(\{\mathbf{1}\})^\perp$.*

(Proof in SM, Section D) A consequence of Theorem 2 is the following Corollary.

**Corollary 5** *For any $\mathsf{A}$, $\mathsf{V}$ such that $\mathsf{A}^\intercal \mathsf{V}\mathbf{1} = \mathbf{0}$, we have*

$$\ell_{\text{CoDA-PCA}}(\mathsf{X}; \mathsf{A}, \mathsf{V}) \leq \tilde{\ell} \doteq \sum_i \frac{1}{g(\boldsymbol{x}_i)} \cdot D_{\check{\text{KL}}}(\boldsymbol{x}_i \,\|\, \exp(\mathsf{V}^\intercal \boldsymbol{a}_i)). \tag{14}$$

*Hence, the CoDA PCA loss is upperbounded by a weighted generalized $\varphi$-PCA loss. Furthermore,*

$$D_{\check{\text{KL}}}(\boldsymbol{x}_i \,\|\, \exp(\mathsf{V}^\intercal \boldsymbol{a}_i)) = \check{\text{KL}}(\boldsymbol{x}_i) - \boldsymbol{x}_i^\intercal \nabla \check{\text{KL}}(\exp(\mathsf{V}^\intercal \boldsymbol{a}_i)) \tag{15}$$

**Proof** Since $g$ is concave (Example 1 in Supplement), $D_g(\boldsymbol{x}\|\boldsymbol{y}) = -D_{-g}(\boldsymbol{x}\|\boldsymbol{y}) \leq 0$, and (14) follows from Theorem 7 and the fact that $r_i \geq 0, \forall i$, which shows (14). (15) is a consequence of the analytical construct of Bregman divergences (Definition 1) and point (3) in Lemma 4 and the fact that $\mathsf{V}^\intercal \boldsymbol{a}_i \in \text{span}(\{\mathbf{1}\})^\perp$ by assumption. ∎

**Remark** In [3, Chapter 8], CoDA PCA is presented as a (centered) regular PCA over data that been subject to *two* transforms via the centered log-ratio coordinates. What Corollary 5 shows is that we can solve the problem via a surrogate formulation using *non transformed data* but minimizing a loss which is that of a $\varphi$-PCA transformed *twice*: first taking a perspective transform of the KL generator (ǨL) and then having a weighted Bregman divergence minimization ($g^{-1}(.)$). We remark that weights can also be folded in the arguments as we have:

$$\tilde{\ell} \;=\; \sum_i \mathrm{KL}(\check{\boldsymbol{x}}_i) - \check{\boldsymbol{x}}_i^{\mathsf{T}} \nabla\check{\mathrm{KL}}(\exp(\mathsf{V}^{\mathsf{T}}\boldsymbol{a}_i)) \;. \tag{16}$$

$\square$

Furthermore, the leftmost argument in (16) plays no role in its minimization, and therefore we get the Surrogate CoDA PCA (S-CoDA-PCA) by replacing (11) with a simple inner product:

$$\ell_{\text{s-CoDA-PCA}}(\mathsf{X};\mathsf{A},\mathsf{V}) \;\doteq\; -\sum_i \check{\boldsymbol{x}}_i^{\mathsf{T}} \nabla\check{\mathrm{KL}}(\exp(\mathsf{V}^{\mathsf{T}}\boldsymbol{a}_i)) \;. \tag{17}$$

## 5   Implementations

Both the CoDA-PCA in (11) and the S-CoDA-PCA in (17) can be equivalently written as the following unconstrained problems

$$(\text{CoDA-PCA}) \quad \underset{\mathsf{B},\mathsf{U}}{\operatorname{argmin}} \; \left[ \mathbf{1}_d^{\mathsf{T}} \exp(\mathsf{Y})\mathbf{1}_m - \operatorname{trace}\left(\check{\mathsf{X}}^{\mathsf{T}}\mathsf{Y}\right) \right], \tag{18}$$

$$(\text{s-CoDA-PCA}) \quad \underset{\mathsf{B},\mathsf{U}}{\operatorname{argmin}} \; \operatorname{trace}\left[ \check{\mathsf{X}}^{\mathsf{T}} \left( \exp(-\mathsf{Y}) \circ \frac{\mathbf{1}_d \mathbf{1}_d^{\mathsf{T}}}{d} \exp(\mathsf{Y}) - \mathsf{Y} \right) \right], \tag{19}$$

where "$\circ$" means element-wise product, $\exp(\cdot)$ is element-wise exponential, and $\mathsf{Y} = \mathsf{C}\mathsf{U}^{\mathsf{T}}\mathsf{B}$ with $\mathsf{C} \in \Re^{d\times d}$, $\mathsf{U} \in \Re^{\ell\times d}$, $\mathsf{B} \in \Re^{\ell\times m}$. $\mathsf{C}$ is a constant centering matrix satisfying $\operatorname{rank}(\mathsf{C}) = d-1$, $\mathsf{C}^{\mathsf{T}}\mathbf{1} = \mathbf{0}$, so that $\mathsf{Y}$'s columns are automatically centered and $\mathsf{Y}^{\mathsf{T}}\mathbf{1} = \mathsf{B}^{\mathsf{T}}\mathsf{U}\mathsf{C}^{\mathsf{T}}\mathbf{1} = \mathbf{0}$. Any $\mathsf{C}$ satisfying this condition corresponds to a valid re-parametrization of the feasible space, for example $\mathsf{C} = \mathsf{I}_d - \frac{1}{d}\mathbf{1}_d\mathbf{1}_d^{\mathsf{T}}$ or $\mathsf{C} = \mathsf{I}_d - \mathsf{I}_d^{\sharp}$ ($\mathsf{I}_d^{\sharp}$ circularly raises the diagonal entries of $\mathsf{I}_d$ by 1 row). $\mathsf{U}$'s rows form a nonorthogonal basis of $\Re^d$. $\mathsf{B}$'s columns are the sample coordinates in such a basis. After optimization, we take the QR decomposition $\mathsf{C}(\mathsf{U}^{\star})^{\mathsf{T}} = (\mathsf{V}^{\star})^{\mathsf{T}}\mathsf{T}^{\star}$, where $\mathsf{V}^{\star}$'s rows are orthonormal. Therefore $\mathsf{C}(\mathsf{U}^{\star})^{\mathsf{T}}\mathsf{B}^{\star} = (\mathsf{V}^{\star})^{\mathsf{T}}\mathsf{T}^{\star}\mathsf{B}^{\star}$ and $\mathsf{A}^{\star} = \mathsf{T}^{\star}\mathsf{B}^{\star}$ is the corresponding coordinates. An optimal solution of the original constrained PCA problem is given by $(\mathsf{V}^{\star}, \mathsf{A}^{\star})$.

Although the losses in (18) and (19) are non-convex, they are both *bi-convex*. Fixing $\mathsf{U}$, the loss is a strictly convex function of $\mathsf{B}$ that is decomposed into a sum of per-sample convex functions of $\boldsymbol{b}_i$; fixing $\mathsf{B}$, it is a strictly convex function of $\mathsf{U}$. These convex functions have the general form $f(\boldsymbol{\xi}) = \sum_i \exp(\boldsymbol{\alpha}_i^{\mathsf{T}}\boldsymbol{\xi} + \beta_i) + \boldsymbol{\zeta}^{\mathsf{T}}\boldsymbol{\xi}$. Its gradient and Hessian are both in simple closed form: $\bigtriangledown f = \sum_i \exp(\boldsymbol{\alpha}_i^{\mathsf{T}}\boldsymbol{\xi}+\beta_i)\boldsymbol{\alpha}_i + \boldsymbol{\zeta}$; $\bigtriangledown^2 f = \sum_i \exp(2\boldsymbol{\alpha}_i^{\mathsf{T}}\boldsymbol{\xi}+2\beta_i)\boldsymbol{\alpha}_i\boldsymbol{\alpha}_i^{\mathsf{T}}$. One can apply an off-the-shelf convex optimizer, which in the simplest case can be the Newton method, to alternately minimize $\mathsf{B}$ and $\mathsf{U}$ until convergence. Our implementation simply uses L-BFGS [9] based on the gradient of the loss. In summary, we have the following result.

**Proposition 6** *The* CoDA-PCA *and the* S-CoDA-PCA *are both equivalent to an unconstrained bi-convex optimization problem.*

As an alternative implementation, we assume a parametric mapping $\boldsymbol{b}_i = g_{\Theta}(\boldsymbol{x}_i)$ that is the $\ell$-dimensional output of a feed-forward neural network with input $\boldsymbol{c}_{\text{KL}}(\boldsymbol{x}_i)$ and $\boldsymbol{x}_i$ (or $\check{\boldsymbol{x}}_i$) and connection weights $\Theta$. Then we minimize the cost function in (18) with respect to $\mathsf{U}$ and $\Theta$. If $g_{\Theta}$ is flexible enough, then the minimization recovers the CoDA-PCA projection. This approach could be favored as ① it learns an out-of-sample mapping $g_{\Theta}(\cdot)$ with a compact parametric structure that does not scale with the sample size $m$; and ② it can be adapted to an online learning scenario. However, it requires tuning of the neural network architecture and the optimizer. In our experiments, the encoding map is modeled by a feed-forward neural network with two hidden layers of ELU [11] units, each of size 100. To distinguish between the two implementations, the method to directly optimize $\mathsf{U}$ and $\mathsf{B}$

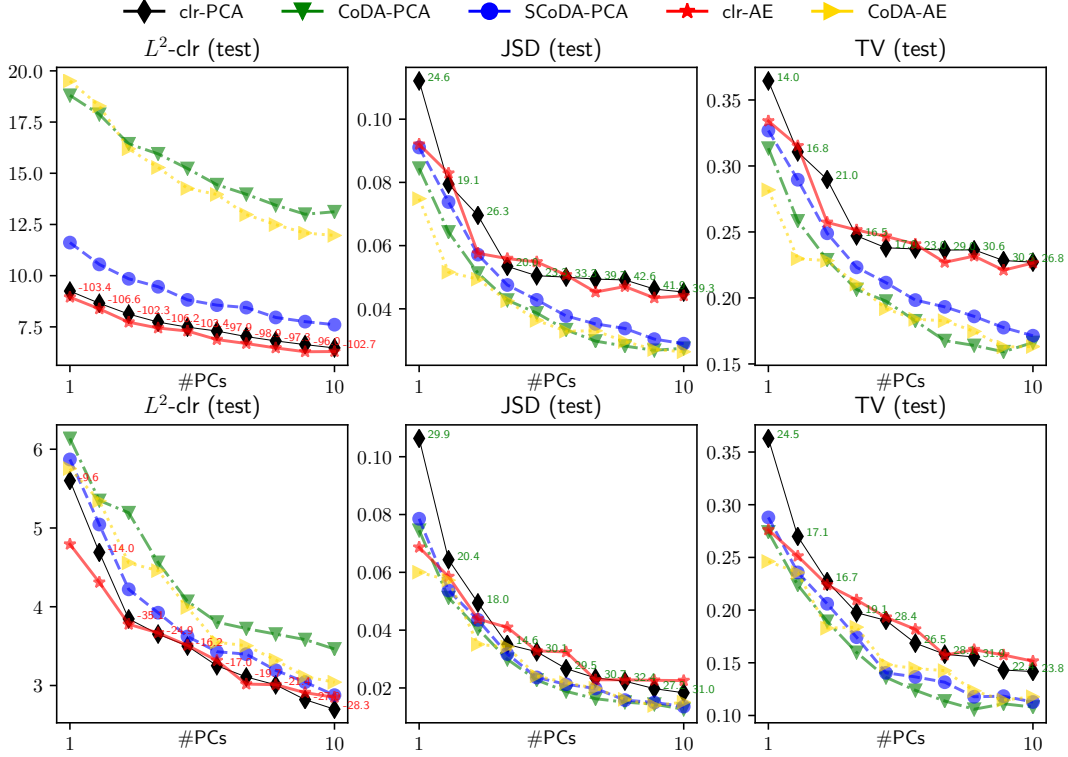

Figure 2: Testing errors (y-axis) against the number of principal components (x-axis) based on three different distance measures (from left to right) on the Atlas data (first row) and the diet swap data (second row). The numbers along the `clr-PCA` curves show the percentage of improvement (green) or disimprovement (red), comparing CoDA-PCA against `clr-PCA`.

without assuming the neural network mapping is called non-parametric CODA-PCA, and the latter parametric version is simply called CODA-PCA.

The above implementation resembles an auto-encoder structure: $\boldsymbol{x}_i \xrightarrow{\Theta} \boldsymbol{b}_i \xrightarrow{\mathsf{U}} \boldsymbol{y}_i$, where the decoder is simply a linear mapping $\boldsymbol{y}_i = \mathsf{CU}^\mathsf{T}\boldsymbol{b}_i$. In the general case, we apply a non-linear decoder $\boldsymbol{b}_i \xrightarrow{\Phi} \boldsymbol{y}_i$ in the form $\boldsymbol{y}_i = \mathsf{C}h_\Phi(\boldsymbol{b}_i)$, where $h_\Phi(\cdot)$ is a neural network with parameters $\Phi$ and $d$ output dimensions. At the same time, we add a small random noise to the encoder input so as to avoid overfitting. In this way we obtain a denoising CODA-AUTOENCODER. In contrast to the CODA-PCA, the CODA-AUTOENCODER can only be trained by gradient-based optimizers.

In practice, the input matrix $\mathsf{X}$ may contain zeros that lie on the boundary of $\mathfrak{S}^d$. In this case $\boldsymbol{c}_{\mathrm{KL}}(\boldsymbol{x})$ and $\check{\boldsymbol{x}}$ are undefined. A simple way to tackle the zero entries is to replace them with a small positive number $\epsilon > 0$. Alternatively, one can redefine the gauge as $g(\boldsymbol{x}) = \prod_{j:x_j>0}(x_j)^{1/\rho}$, where $\rho = |\{j : x_j > 0\}|$ so that $g(\boldsymbol{x})$ is always positive and $\check{\boldsymbol{x}}$ is well defined on $\mathfrak{S}^d \cup \partial\mathfrak{S}^d$.

## 6   Experiments

We compare the following methods: `clr-PCA` means PCA applied on the centered log-ratio coordinates; `CoDA-PCA` is the proposed CODA-PCA in (11); `SCoDA-PCA` is the proposed S-CODA-PCA in (17); `clr-AE` is an autoencoder with $L^2$ loss applied on the clr transformation; `CoDA-AE` is the proposed CODA-AUTOENCODER in subsection 5. Both `clr-AE` and `CoDA-AE` use exactly the same structure with one hidden layer of 100 ELU [11] units in their decoders.

The baselines are assessed based on an array of measures including ($L^2$-clr) the $L^2$-distance $\|\boldsymbol{c}_{\mathrm{KL}}(\boldsymbol{x}) - \boldsymbol{c}_{\mathrm{KL}}(\boldsymbol{x}')\|_{\mathrm{F}}$ between the input data $\boldsymbol{x} \in \mathfrak{S}^d$ and the reconstruction $\boldsymbol{x}' \in \mathfrak{S}^d$ in the clr space; (JSD) the Jensen-Shannon divergence $\frac{1}{2}\mathrm{KL}(\boldsymbol{x} : \frac{\boldsymbol{x}+\boldsymbol{x}'}{2}) + \frac{1}{2}\mathrm{KL}(\boldsymbol{x}' : \frac{\boldsymbol{x}+\boldsymbol{x}'}{2})$; (TV) the total variation distance

$\frac{1}{2} \sum_{i=1}^{d} |x_i - x_i'|$. These measurements are all invariant to scaling or permutation of $x$ and $x'$. See the supplementary material for more baselines and performance indicators.

We consider the following datasets available in the microbiome R package [18], each of which is randomly split into a training set (90%) and a testing set (10%). *The HITChip Atlas dataset [17]* contains 130 genus-level taxonomic groups that cover the majority of the known bacterial diversity of the human intestine. The data come from 1006 western adults from 15 western countries (Europe and the United States). Sample sets were analysed with three different DNA extraction methods. *The two-week diet swap study* between western (USA) and traditional (rural Africa) diets was reported in [28]. In this study, a two-week food exchange was performed in subjects from the same populations, where African Americans were fed a high-fibre, low-fat African-style diet and rural Africans a high-fat, low-fibre western-style diet. The group diet was indicated by HE (home environment days), DI (dietary intervention days) and ED (initial and final endoscopy days). Each subject served as his/her own control, given the known wide individual variation in colonic microbiota composition.

Fig. 2 shows the typical testing results. We observe that on most performance indicators `CoDA-PCA` and `CoDA-AE` show a much smaller testing error as compared to `clr-PCA` and `clr-AE`, respectively. The only exception is on $L^2$-clr, where `clr-PCA` and `clr-AE` appear to be favored against our `CoDA` variants. This is because $L^2$-clr is exactly the cost function of those two methods. We found that `CoDA-AE` is more robust against overfitting as compared to `clr-AE`. The performance of `SCoDA-PCA` is close to `CoDA-PCA` on most of the indicators and is better than `CoDA-PCA` on $L^2$-clr.

The source codes to reproduce our experimental results are available online[2].

# 7   Conclusion

We propose an approach for learning a low dimensional representation directly on raw count data, which is compositional in nature. Our proposed algorithm generalizes PCA in two ways, first by going to the exponential family via the Bregman divergence, and second by converting the normalization of data to a change in the Bregman divergence. The key theorem used for transforming the Bregman divergence generalizes a recent result, and may be of independent interest.

**Acknowledgements**

The authors gratefully thank Perrine Soret, Frank Nielsen, Xinhua Zhang, and the anonymous NIPS reviewers, for their helpful and constructive feedback. This work was done while MAF was visiting Data61, CSIRO in Canberra, Australia.

## Footnotes

*Authors in alphabetical order

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
