[Supplementary Material]

# A Convexity of divergence

**Theorem 7** *For any* A, V *such that* $A^\intercal V1 = 0$, *letting* $D_{\exp}$ *the Bregman divergence with generator* $\varphi^\star(z) \doteq \exp z$ *and* KL *the generator for KL divergence,*

$$D_{\exp}(V^\intercal a_i \| c_{\mathrm{KL}}(x_i)) = q_i \cdot D_{\mathrm{\check{K}L}}(x_i \| \exp(V^\intercal a_i)) + r_i \cdot D_g(x_i \| \exp(V^\intercal a_i)), \forall i = 1, 2, ..., m \quad (20)$$

*with* $q_i \doteq 1/g(x_i)$ *and* $r_i \doteq q_i \cdot \sum_j \exp(a_i^\intercal v_j)$, *satisfying* $r_i \geq dq_i$.

(Proof in SM, Section B)

We can remark that

$$\frac{1}{g(x_i)} \cdot \sum_j \exp(a_i^\intercal v_j) = \sum_j \exp(a_i^\intercal v_j - x_{ij}) \check{x}_{ij}. \quad (21)$$

To generalize the notion of geometric average, consider any strictly convex function $\varphi : \mathbb{R} \to \mathbb{R}$ at least three times differentiable and with invertible derivative. Define the $\varphi$-mean of $x$ as

$$\mu_\varphi(x) \doteq \varphi'^{-1}(\mathsf{E}_i \varphi'(x_i)). \quad (22)$$

We now state a Lemma which will be essentially used for a particular case of $\varphi$ but may be of independent interest for more general cases, as explained in the following examples.

**Lemma 8** *Let* $\varphi$ *convex and at least three times differentiable. Let*

$$\phi(x) \doteq -\frac{\varphi'''(x)}{(\varphi''(x))^2}. \quad (23)$$

*Then the* $\varphi'$-*mean* $\mu_\varphi$ *is convex (resp. concave) iff*

$$\phi(\mu_\varphi(x)) \geq (resp. \leq) \quad m \cdot \min_i \phi(x_i), \forall x. \quad (24)$$

(Proof in SM, Section C)

**Example 1** *Take for example* $F'(x) \doteq \ln x$ *(geometric average). In this case,* $\phi(x) = +1$ *and ineq. (24) brings* $1 \leq m$ *for the concavity of the mean and shows that g is concave.*

**Example 2** *Consider* $F'(x) = -1/x$ *(harmonic average, over* $\mathbb{R}_+$*). In this case,* $\phi(x) = +x$, *so to prove the concavity of the mean, we need to show* $\mu_F(x) \leq m \min_i x_i$, *which is equivalent to show*

$$\sum_i \frac{1}{x_i} \geq \max_i \frac{1}{x_i}, \quad (25)$$

*and shows the concavity of the mean.*

**Example 3** *Consider* $F' = \exp$ *(normalized softmax). In this case,* $\phi(x) = -\exp(-x)$, *which shows that the mean cannot be concave. To show its convexity, we need to show equivalently*

$$\frac{1}{\sum_i \exp x_i} \leq \exp(-\max_i x_i), \quad (26)$$

*which is equivalent to* $\sum_i \exp(x_i - \max_j x_j) \geq 1$, *and because of the non-negativity of the exponential, shows the convexity of the mean.*

# B Proof of Theorem 7

First, letting $y_i \doteq \exp(V^\intercal a_i)$ for any $i = 1, 2, ..., m$, and $v_j$ the $j$-th row of V (as a column vector), we get

$$\varphi^\star(\nabla\varphi(\check{y}_i)) = \sum_j \exp(\log(\exp(a_i^\intercal v_j))) = \sum_j \exp(a_i^\intercal v_j). \quad (27)$$

We recall $g(\boldsymbol{x}) = (\prod_k x_k)^{1/d}$, the geometric mean. So,

$$g(\exp(\mathsf{V}^\mathsf{T}\boldsymbol{a}_i)) \;\; = \;\; \exp\left(\frac{1}{d}\sum_j \boldsymbol{a}_i^\mathsf{T}\boldsymbol{v}_j\right) = \exp((\mathsf{A}^\mathsf{T}\mathsf{V}\mathbf{1})_i) = \exp(0) = 1, \tag{28}$$

because of the constraint $\mathsf{A}^\mathsf{T}\mathsf{V}\mathbf{1} = \mathbf{0}_m$. Hence, letting $\boldsymbol{y}_i \doteq \exp(\mathsf{V}^\mathsf{T}\boldsymbol{a}_i)$ for short, we obtain $\check{\boldsymbol{y}}_i = \boldsymbol{y}_i$. Also, the dual symmetry of Bregman divergences [8] yields:

$$D_{\exp}(\mathsf{V}^\mathsf{T}\boldsymbol{a}_i \| \boldsymbol{c}_{\mathrm{KL}}(\boldsymbol{x}_i)) \;\; = \;\; D_{\mathrm{KL}}(\check{\boldsymbol{x}}_i \| \exp(\mathsf{V}^\mathsf{T}\boldsymbol{a}_i)) \; . \tag{29}$$

We now use Theorem 2. It comes (letting again $\boldsymbol{y}_i \doteq \exp(\mathsf{V}^\mathsf{T}\boldsymbol{a}_i)$ for short) that for any $i = 1, 2, ..., m$,

$$
\begin{aligned}
&g(\boldsymbol{x}_i) \cdot D_{\exp}(\mathsf{V}^\mathsf{T}\boldsymbol{a}_i \| \boldsymbol{c}_{\mathrm{KL}}(\boldsymbol{x}_i)) \\
&= g(\boldsymbol{x}_i) \cdot D_{\mathrm{KL}}(\check{\boldsymbol{x}}_i \| \exp(\mathsf{V}^\mathsf{T}\boldsymbol{a}_i)) \\
&= D_{\check{\mathrm{KL}}}\left(\boldsymbol{x}_i \;\middle\|\; \exp(\mathsf{V}^\mathsf{T}\boldsymbol{a}_i)\right) + \varphi^\star\left(\nabla\varphi(\check{\boldsymbol{y}}_i)\right) D_g(\boldsymbol{x}_i \| \boldsymbol{y}_i) \\
&= D_{\check{\mathrm{KL}}}\left(\boldsymbol{x}_i \;\middle\|\; \exp(\mathsf{V}^\mathsf{T}\boldsymbol{a}_i)\right) + \left(\sum_j \exp\left(\boldsymbol{a}_i^\mathsf{T}\boldsymbol{v}_j\right)\right) \cdot D_g(\boldsymbol{x}_i \| \exp(\mathsf{V}^\mathsf{T}\boldsymbol{a}_i)) \;,
\end{aligned}
\tag{30}
$$

which brings the statement of the Theorem. The fact that $r_i \geq dq_i$ follows from the convexity of $\exp$.

## C   Proof of Lemma 8

Let $\mu_\varphi(\boldsymbol{x}) \doteq \varphi'^{-1}((1/m) \cdot \sum_i \varphi'(x_i))$ the $\varphi'$-mean of the coordinates of $\boldsymbol{x}$. Then

$$\frac{\partial}{\partial x_i}\mu_\varphi(\boldsymbol{x}) \;\; = \;\; \frac{1}{m}\cdot\frac{\varphi''(x_i)}{\varphi''(\mu_\varphi(\boldsymbol{x}))} \;, \tag{31}$$

$$\frac{\partial^2}{\partial x_i \partial x_j}\mu_\varphi(\boldsymbol{x}) \;\; = \;\; \frac{1}{m\varphi''(\mu_\varphi(\boldsymbol{x}))}\cdot\left(\delta_{ij}\cdot\varphi'''(x_i) - \frac{\varphi'''(\mu_\varphi(\boldsymbol{x}))}{m(\varphi''(\mu_\varphi(\boldsymbol{x})))^2}\cdot\varphi''(x_i)\varphi''(x_j)\right) \tag{32}$$

$\boldsymbol{x}$ being fixed, let $\tilde{\boldsymbol{y}}$ the vector defined by $\tilde{y}_i \doteq y_i \cdot \varphi''(x_i)$ and let

$$\phi(x) \;\; \doteq \;\; -\frac{\varphi'''(x)}{(\varphi''(x))^2} \;. \tag{33}$$

Let $\Phi$ the diagonal matrix with $\Phi_{ii} \doteq \phi(x_i)$. The Hessian $\mathrm{H}_{\mu_\varphi(\boldsymbol{x})}$ satisfies

$$
\begin{aligned}
\boldsymbol{y}^\mathsf{T}\mathrm{H}_{\mu_\varphi(\boldsymbol{x})}\boldsymbol{y} \;\; &= \;\; \frac{1}{m\varphi''(\mu_\varphi(\boldsymbol{x}))}\cdot\left(\frac{\phi(\mu_\varphi(\boldsymbol{x}))}{m}\cdot\tilde{\boldsymbol{y}}^\mathsf{T}\tilde{\boldsymbol{y}} - \tilde{\boldsymbol{y}}^\mathsf{T}\Phi\tilde{\boldsymbol{y}}\right) \\
&= \;\; \frac{1}{m\varphi''(\mu_\varphi(\boldsymbol{x}))}\cdot\tilde{\boldsymbol{y}}^\mathsf{T}\mathrm{Diag}\left(\frac{\phi(\mu_\varphi(\boldsymbol{x}))}{m}\cdot\mathbf{1} - \phi(\boldsymbol{x})\right)\tilde{\boldsymbol{y}}
\end{aligned}
\tag{34}
$$

We see that when $\varphi''$ is not zero everywhere, $\tilde{\boldsymbol{y}}$ spans the same set as $\boldsymbol{y}$, so if $\varphi$ is convex, $\mu_\varphi$ is convex iff $(\phi(\mu_\varphi(\boldsymbol{x}))/m) \cdot \mathbf{1} - \phi(\boldsymbol{x}) \geq \mathbf{0}$, that is

$$\phi(\mu_\varphi(\boldsymbol{x})) \;\; \geq \;\; m \cdot \phi(x_i) \;, \forall i \;, \tag{35}$$

which is equivalent to the Lemma's statement.

## D   Proof of Lemma 4

We have

$$\check{\mathrm{KL}}(\boldsymbol{x}) \;=\; \left(\sum_i x_i \log x_i - x_i\right) - \frac{1}{d}\cdot\sum_{ij} x_i \log x_j \;, \tag{36}$$

$$\frac{\partial}{\partial x_i}\check{\mathrm{KL}}(\boldsymbol{x}) \;=\; \log x_i - \frac{1}{d}\cdot\sum_j \log x_j - \frac{1}{dx_i}\cdot\sum_j x_j \tag{37}$$

$$\;=\; -\frac{1}{d}\cdot\left(\sum_j \frac{x_j}{x_i} + \log\frac{x_j}{x_i}\right)\;, \tag{38}$$

$$\frac{\partial^2}{\partial x_i \partial x_j}\check{\mathrm{KL}}(\boldsymbol{x}) \;=\; -\frac{1}{d}\cdot\left(\frac{1}{x_i} + \frac{1}{x_j}\right), \forall j \neq i, \tag{39}$$

$$\frac{\partial^2}{\partial x_i^2}\check{\mathrm{KL}}(\boldsymbol{x}) \;=\; \frac{1}{x_i}\cdot\left(1 - \frac{1}{d} + \frac{1}{d}\cdot\sum_{j\neq i}\frac{x_j}{x_i}\right). \tag{40}$$

We then check that

$$(\mathrm{H}\boldsymbol{z})_i \;=\; \frac{z_i}{x_i} + \frac{1}{d}\cdot\sum_j\left(\frac{z_i x_j}{x_i^2} - \frac{z_j}{x_j} - \frac{z_j}{x_i}\right) \tag{41}$$

and finally

$$\begin{aligned}
\boldsymbol{z}^{\mathsf{T}}\mathrm{H}\boldsymbol{z} &\;=\; \sum_i \frac{z_i^2}{x_i} + \frac{1}{d}\sum_{ij}\left(\frac{z_i^2 x_j}{x_i^2} - \frac{z_i z_j}{x_j} - \frac{z_i z_j}{x_i}\right) \\
&\;=\; \frac{1}{d}\sum_{ij}\left(\frac{z_i^2 x_i}{x_i^2} + \frac{z_i^2 x_j}{x_i^2} - \frac{z_i z_j}{x_j} - \frac{z_i z_j}{x_i}\right) \\
&\;=\; \frac{1}{d}\sum_{ij}\left(\frac{z_i^2 x_i x_j^2 + z_i^2 x_j^3 - z_i z_j x_i^2 x_j - z_i z_j x_i x_j^2}{x_i^2 x_j^2}\right) \\
&\;=\; \frac{1}{2d}\sum_{ij}\left(\frac{z_i^2 x_i x_j^2 + z_i^2 x_j^3 + z_j^2 x_i^2 x_j + z_j^2 x_i^3 - 2z_i z_j x_i^2 x_j - 2z_i z_j x_i x_j^2}{x_i^2 x_j^2}\right) \\
&\;=\; \frac{1}{2d}\sum_{ij}\left(\frac{z_i^2 x_j^2(x_i + x_j) + z_j^2 x_i^2(x_i + x_j) - 2z_i z_j x_i x_j(x_i + x_j)}{x_i^2 x_j^2}\right) \\
&\;=\; \frac{1}{2d}\sum_{ij}(x_i + x_j)\cdot\left(\frac{z_i}{x_i} - \frac{z_j}{x_j}\right)^2\;,
\end{aligned} \tag{42}$$

as claimed. This also shows the convexity of $\check{\mathrm{KL}}(\boldsymbol{x})$.

We now show that $\check{\mathrm{KL}}\circ\exp$ is 1-homogeneous on the subspace $\mathrm{span}(\{\mathbf{1}\})^\perp$, which means, by Euler's Theorem,

$$\check{\mathrm{KL}}(\exp(\boldsymbol{x})) \;=\; \exp(\boldsymbol{x})^{\mathsf{T}}\nabla\check{\mathrm{KL}}(\exp(\boldsymbol{x})), \forall \boldsymbol{x}\in\mathrm{span}(\{\mathbf{1}\})^\perp. \tag{43}$$

To see this, we write

$$
\begin{aligned}
&\check{\mathrm{KL}}(\exp(\boldsymbol{x})) - \exp(\boldsymbol{x})^{\mathsf{T}}\nabla\check{\mathrm{KL}}(\exp(\boldsymbol{x})) \\
&= \sum_j x_j \exp(x_j) - \exp(x_j) \\
&\quad + \frac{1}{d}\cdot\sum_j\left(\exp(x_j)\cdot\sum_k\left(\exp(x_k - x_j) + (x_k - x_j)\right)\right) \\
&= \sum_j x_j \exp(x_j) \underbrace{- \sum_j \exp(x_j) + \sum_k \exp(x_k)}_{=0} \\
&\quad + \frac{1}{d}\cdot\sum_j \exp(x_j)\underbrace{\sum_k x_k}_{=0} - \sum_j \exp(x_j)x_j \\
&= \sum_j x_j \exp(x_j) - \sum_j \exp(x_j)x_j \\
&= 0,
\end{aligned}
\tag{44}
$$

where we have used the fact that $\mathbf{1}^{\mathsf{T}}\boldsymbol{x} = 0$.

## E    Derivations of the unconstrained optimization in (18) and (19)

Consider that $\mathsf{X}$ is constant, therefore

$$
\begin{aligned}
\ell_{\mathrm{CoDA\text{-}PCA}}(\mathsf{X};\mathsf{A},\mathsf{V}) &= D_{\exp}(\mathsf{V}^{\mathsf{T}}\mathsf{A}\,\|\,\boldsymbol{c}_{\mathrm{KL}}(\mathsf{X})) \\
&= \mathbf{1}^{\mathsf{T}}\left[\exp(\mathsf{V}^{\mathsf{T}}\mathsf{A}) - \exp(\boldsymbol{c}_{\mathrm{KL}}(\mathsf{X})) - (\mathsf{V}^{\mathsf{T}}\mathsf{A} - \boldsymbol{c}_{\mathrm{KL}}(\mathsf{X}))\circ\exp(\boldsymbol{c}_{\mathrm{KL}}(\mathsf{X}))\right]\mathbf{1} \\
&= \mathbf{1}^{\mathsf{T}}\left[\exp(\mathsf{V}^{\mathsf{T}}\mathsf{A}) - \mathsf{V}^{\mathsf{T}}\mathsf{A}\circ\exp(\boldsymbol{c}_{\mathrm{KL}}(\mathsf{X}))\right]\mathbf{1} + \text{constant} \\
&= \mathbf{1}^{\mathsf{T}}\left[\exp(\mathsf{V}^{\mathsf{T}}\mathsf{A}) - \mathsf{V}^{\mathsf{T}}\mathsf{A}\circ\check{\mathsf{X}}\right]\mathbf{1} + \text{constant}.
\end{aligned}
$$

Let $\mathsf{Y} = \mathsf{V}^{\mathsf{T}}\mathsf{A}$, and we get (18).

By (38), if $\boldsymbol{y}_i$ is centered and $\boldsymbol{y}_i^{\mathsf{T}}\mathbf{1} = 0$, we have

$$
\begin{aligned}
\nabla\check{\mathrm{KL}}(\exp(\boldsymbol{y}_i)) &= -\frac{1}{d}\cdot\left[\exp(\mathbf{1}\boldsymbol{y}_i^{\mathsf{T}} - \boldsymbol{y}_i\mathbf{1}^{\mathsf{T}}) + \mathbf{1}\boldsymbol{y}_i^{\mathsf{T}} - \boldsymbol{y}_i\mathbf{1}^{\mathsf{T}}\right]\mathbf{1} \\
&= -\frac{1}{d}\exp(\mathbf{1}\boldsymbol{y}_i^{\mathsf{T}} - \boldsymbol{y}_i\mathbf{1}^{\mathsf{T}})\mathbf{1} + \boldsymbol{y}_i.
\end{aligned}
$$

Therefore

$$
\begin{aligned}
\ell_{\mathrm{s\text{-}CoDA\text{-}PCA}}(\mathsf{X};\mathsf{A},\mathsf{V}) &= \sum_i\left[\check{\boldsymbol{x}}_i^{\mathsf{T}}\frac{1}{d}\exp(\mathbf{1}\boldsymbol{y}_i^{\mathsf{T}} - \boldsymbol{y}_i\mathbf{1}^{\mathsf{T}})\mathbf{1} - \check{\boldsymbol{x}}_i^{\mathsf{T}}\boldsymbol{y}_i\right] \\
&= \sum_i\check{\boldsymbol{x}}_i^{\mathsf{T}}\left[\exp(-\boldsymbol{y}_i)\exp(\boldsymbol{y}_i^{\mathsf{T}})\frac{\mathbf{1}}{d} - \boldsymbol{y}_i\right],
\end{aligned}
$$

which, in matrix form, is (19).

## F    More Experimental Results

We include another baseline CoDA-PCA$^*$ that is the non-parametric version of CoDA-PCA optimized by L-BFGS. Note that CoDA-PCA$^*$ does not learn an encoding map and therefore cannot provide an out-of-sample extension.

The baselines are further assessed based on (symmetric perspective KL divergence; SPKL) normalizing the geometric average of the input data $p$ and the PCA reconstruction $q$ and get respectively two positive measures $\check{p}$ and $\check{q}$, then computing the symmetric KL divergence $\frac{1}{2} \sum_i \left( \check{p}_i \log \frac{\check{p}_i}{\check{q}_i} + \check{q}_i \log \frac{\check{q}_i}{\check{p}_i} \right)$; ($L^2$) $L^2$-distance of $\bar{p}$ and $\bar{q}$ after the normalizing $p$ and $q$ into the probability simplex. (Riemannian) the Riemannian distance between the two probabilities $\bar{p}$ and $\bar{q}$ defined by the Fisher information metric, given by $2 \arccos \left( \sqrt{\bar{p}}^{\mathsf{T}} \sqrt{\bar{q}} \right)$.

Fig. 3 shows the training and testing errors. A key observation is that CoDA-PCA* is slightly better than CoDA-PCA because the embedding points are free parameters and are not constrained by a neural network. We also see that clr-AE shows small training errors but does not generalize as well as CoDA-AE on the testing set.

Figure 3: Training and testing errors measured by different distances against the number of principal components. The columns, from left to right, show training errors (Atlas), corresponding testing errors, training errors (diet swap) and corresponding test errors.