[Reviews · NeurIPS 2018]

Reviewer 1



This paper suggests a generalization of PCA that is applicable to compositional data (non-negative data where each sample sums to 1 or to 100%). It builds up on exponential family PCA developed in the early 2000s. There have been multiple more or less ad hoc approaches of transforming compositional data to make it amenable to PCA, and a more principled method of compositional PCA would certainly be an interesting development. The paper is reasonably well written and uses interesting mathematics. However, it (a) lacks convincing examples, (b) is somewhat confusing in places. I put it above the acceptance threshold, hoping that the authors would be able to improve the presentation. Caveat: I am not familiar with Bregman divergences and do not know the [22] paper that this one heavily uses. I cannot therefore judge on the details of the math. MAJOR ISSUES 1. When reading the paper, I was waiting for some cool visualizations of real-life data. I was disappointed to find *no* visualizations in Section 5.2. Actually the text says "visualizations of the two main axes..." (line 233 and 246), but they are not shown!! Please show! Also, you seem to be saying that all methods gave similar results (e.g. line 236). If so, what's the point of using your method over more standard/simple ones? I would like to see some example where you can show that your method outperforms vanilla PCA and alr/clr/ilr+PCA in terms of visualization. Your metrics in Figure 2 are not very convincing because it's hard to judge which of them should be most relevant (maybe you should explain the metrics better). 2. The toy example (Figure 1) is good, but I would like to see alr+PCA and ilr+PCA in addition to clr+PCA (I know that different people recommend different choices; does it matter here?), and more importantly, I'd like to see vanilla PCA. It seems to me that at least for 3-arms vanilla PCA should give an ideal "Mercedes" shape without any log-transforms. Is it true for 10-arms? If it _is_ true, then this is a poor example because vanilla PCA performs optimally. If so, the authors should modify the toy example such that alr/clr/ilr+PCA performed better than PCA (this would also help motivating clr-transform) and that CodaPCA performed even better than that. 3. I think a better motivation is needed for log-transforms in Section 2. Line 52 mentions "nonlinear structure due to the simplex constraint" but simplex constraint is clearly linear. So if the simplex constraint were the only issue, I wouldn't see *any* problem with using standard vanilla PCA. Instead, there arguably *is* some nonlinearity in the nature of compositional data (otherwise why would anybody use log-transforms), and I remember Aitchinson showing some curved scatterplots on the 2D simplex. This needs to be explained/motivated here. 4. Line 91 says that count data can be described with Poisson distribution. Line 260 again mentions "raw count data". But compositional data does not need to be count, it can come as continuous fractions (e.g. concentrations). Does the method suggested here only apply to counts? If so, this should be made very clear in the title/abstract/etc. If not, please clarify why Poisson distribution still makes sense. MINOR ISSUES 1. Box after line 81. Shouldn't Eq 3 be part of the box? Without it the box looks incomplete. The same for the box after line 107. 2. Eq 6. "\phi-PCA" should be written with a hyphen (not minus) and capitalized PCA. Please use hyphens also below in similar expressions. 3. The end of Section 3 -- I suggest to insert some comments on how exactly exp family PCA reduces to standard PCA in the Gaussian case. What is \phi function, what is Bregman divergence, etc. 3. Line 113, last sentence of the section. This is unclear. Why is it a hole? Why is not easy? 4. Line 116, first sentence of the section: again, not clear how exactly clr is a workaround. 5. Line 139: period missing. 6. Line 144, where does this \phi() come from? Is it Poisson? Please explain. 7. Eq (11): what is "gauged-KL", the same as "gauged-phi" above? 8. Box after line 150: now you have loss function in the box, compare with minor issue #1. 9. Lines 201 and 214: two hidden layers or one hidden layer? 10. Line 206: space missing. 11. Line 211: isn't CoDA-PCA and SCoDA-PCA the same thing? Got confused here. 12. Line 257: I don't see any overfitting in Figure 2. This would be test-set curve going upwards, no? Please explain why do you conclude there is overfitting. 13. Shouldn't you cite Aitchinson's "PCA of compositional data" 1983? ----------------- **Post-rebuttal update:** I still think the paper is worth accepting, but my main issue remains the lack of an obviously convincing application. I see that reviewer #3 voiced the same concern. Unfortunately, I cannot say that the scatter plots shown in the author response (panels A and B) settled this issue. The authors write that panel A, unlike B, shows three clusters but to me it looks like wishful thinking. I am not familiar with the microbiome data but I do agree with the point of reviewer #3 that with 100+ features the compositional nature of the data can simply be ignored without losing much. Would the authors agree with that? I think it would be worth discussing. Overall, it looks like a good attempt at setting up composional PCA in a mathematically principled way, but in the end I don't see examples where it performs noticeably better than standard naive approaches.

Reviewer 2



In this paper the authors introduce CoDA-PCA and a surrogate variant for representation learning of compositional data. They introduce and derive their framework on the context of the standard in the field being PCA applied to a transform of the original data that lies on the simplex (here the CLR transform), and demonstrate that their loss function upper bounds the exponential family PCA loss applied to the transformed data. An extension is developed incorporating a deep-neural-net likelihood and the authors apply their methods to several real datasets measuring accuracy in terms of variants of reconstruction error, convincingly demonstrating an improvement over the current standard. Overall the paper seems technically sound and introduces an original method to the field. I think it would be greatly improved if the fundamental message was made clearer (transformed vs untransformed, see point 1 below), and some example visualizations of the datasets were included. Overall I found the paper mathematically very dense with sometimes too short explanations, though I appreciate the 8 page limit imposed. Further comments below. This paper essentially proposes two approaches to PCA for compositional data - a CoDA-PCA model that operates on CLR-transformed count (or otherwise) data, and the surrogate CoDA-PCA that operates on the original count data. The authors allude to the fact that models that operate on untransformed data are advantageous and this is the main proposal of the paper as outlined at the end of the introduction. However, on the experiments in section 5 they demonstrate that their model that operates on the transformed data performs better than the variant that operates on the untransformed. Does this alter their original belief that operating on the original data is advantageous? I’m also curious about the justification of this belief in the first place - I can understand it in the context of generative models and wanting to have calibrated uncertainties, but if one is simply defining a loss function and optimizing, it seems a moot point as to whether the data is transformed or not? Towards the end of the introduction, the authors frame their method as being required to overcome the issue of scaling data as input to PCA. However, depending on the variant of PCA used this isn’t strictly necessary, e.g. in PPCA the data is centred to remove the need to explicitly model an “intercept” term but in practice this can be easily included. Can the authors comment on this in the context of their model, i.e. could an intercept be incorporated? Given the important justification and motivation of compositional PCA for understanding the blossoming field of microbiome studies I think it is important to plot the latent space as fitted to the real data in the paper. Does it appear dramatically different from the existing variants? Does it differentiate or identify e.g. different strains any differently? I somewhat object to the authors’ repeated description of count data as “crude” - can they clarify what they mean by this? Line 65 - the authors explain the drawbacks of the CLR transformation as “car leads to degenerate distributions and singular covariance matrix” - given their subsequent results are essentially based around the CLR transformations, can the authors elaborate on these drawbacks and discuss how they might apply to their proposed model? Line 73 and figure 1 - is the tSNE representation computed using the CLR-transformed data or the original untransformed? The authors refer to the loss of the CoDA-PCA model (equation 7) as the “CoDA loss” (e.g. on line 117), but this is somewhat confusing as pretty much all the losses are “compositional data analysis losses”, so it would be good to stick to the “CoDA-PCA loss” or similar Lines 119-121 state “However, if we wish to perform PCA on the crude count data, while maintaining the clr transform, we need an additional normalization term…”. Can the authors explain this further? Indeed equation 7 is essentially PCA on clr-transformed “crude” count data, so what they state appears already performed? I think it’s important the authors include a version of dimensionality reduction in their real-world experiments (section 5) that doesn’t take into account the compositional nature of the data. This is important as the large difference in log-likelihood achieved would convince the reader of the importance of the proposed compositional methods, and similarly as the data presented contains many features (~130) which in practice may mean the overall denominator is approximately constant and the data doesn’t exhibit compositional “properties” per se. I don’t understand the inset text in section 5.2, e.g. the sentence “Visualization of the two main axes of Principal Component Analysis based on the 130 genus-like groups across the 1006 samples.” This doesn’t seem to reference or be shown as a figure? Is this PCA on the raw data, transformed data, or using methods proposed by the author? A similar sentence is present in the next paragraph also. Line 60 - “loosing”

Reviewer 3



Summary: Recall standard Principal Component Analysis (PCA) with respect to squared norm can be generalized into PCA for exponential family where minimizing Bregman divergence with respect to Legendre conjugate of cumulant functions [5, 9, 17]. This paper first presents this type of generalization of PCA for compositional data, non-negative data with constant sum (data bounded onto a specific simplex). In traditional statistics, handling with compositional data by standard methods is known to be problematic and result in misleading conclusions. Due to developments of Aitchison's Compositional Data Analysis (CoDA), pre-transformation by alr, clr, or ilr would be a standard way to handle compositional data. This paper focus on clr which can be regarded as the form of log(x/g(x)) by gauge g (actually, geometric mean) as seen in eq (1), and thus we can reformulated with a gauged version of Bregman divergence called scaled Bregman divergence. According to this observation, the paper presents a PCA formulation (CoDA-PCA) for clr-transformed data of exponential family in terms of the scaled Bregman divergence. After further relating this CoDA PCA to KL divergence, it turned out that the loss of CoDA PCA is upperbounded in a specific form (weighted generalized phi-PCA in the paper). Minimizing this upper bound leads a rather simple formulation of so-call Surrogate CoDA PCA (s-CoDA-PCA) on the 'original' data before clr transformation. This understanding enables us to further use autoencoder to replace linear coordinate transformation by a nonlinear transformation, and thus provides a method for 'representation learning of compositional data' as the paper's title indicates. The experiments using microbiome data confirms that the proposed method works for these real-world datasets with respect to many possible measures. Strengths: - This paper provides interesting connections of some theoretical ideas such as PCA for exponential families by Bregman divergence and reparametrization by Legendre conjugate of cumulant functions, clr transformation and the gauged version of Bregman divergence (scaled Bregman divergence), derivation of the upperbound of CoDA PCA as well as Surrogate CoDa PCA, and its extension with autoencoders. - In addition to the intuitive visualization as in Figure 1, the experiments of PCA results using microbiome datasets are evaluated by many measures, and on most of the measures proposed CoDA-PCA and CoDA-AE are favored against the others. - Experimental results are based on all of mentioned methods, CLR-PCA and CLR-AE vs CoDA-PCA, SCoDA-PCA, CoDA-AE. Some observations such as the unstable behaviors of CLR-AE against CoDA-AE, and the fact SCoDA-PCA is mostly good against CoDA-PCA are quite interesting. Weakness: - All of this paper's results highly depends on the use of clr for handling compositional data, because it is of the form of a 'gauged' function. But for traditional treatment of Aitchison geometry, ilr rather than clr would be another popular choice. But this paper's formulation seems to be difficult to apply to ilr, and the implications by this generalization might be limited. Actually, I'm not so sure whether applying PCA to clr-transformed data would be a typical option. - Some parts of theoretical notations and concepts are simply undefined or a bit hard to follow. For example, at the line 109, cite [2] (and [17]) for the formulation of eq (5), but is this true? ([2] is the book by Aichison, and might be wrong, [5] or [9] or so?). Another unclear symbol is R_* of the domain of function g. Is this any limitation or a standard notation in some math subfield? - The paper's title is a bit misleading, and it seems to emphasize the CoDA-AE part, but most of the paper are actually devoted to present CoDA-PCA and s-CoDA-PCA, and the 'representation learning' discussion is quite limited. Comment: - Is the formulation of eq (5) well known result? I've heard of the formulation of [9] for PCA for exponential family. But I'm not familiar with introducing the gradient of conjugate in this formulation. Does this come from reparametrization of coordinates in information geometry such as [5] or something? It would be helpful to add some references, or some exact pointers to understand this seemingly more modern formulation than the one [9]. - At line 132, 'conformal divergence are defined and analyzed in [23]', but how about the 'perspective divergence'...? I couldn't understand this abstract statement above the line 132. Comment after author response: Thank you for the response. It's nice to hear that the authors make links clearer between: exponential family PCA, scaled Bregman theorem, and compositional data as well as the careful recheck and polish of the formulations and notations. I also understood, as the authors claim, that PCA can be regarded as linear representation learning, which would be relevant for the paper's title.